# Interventions to promote access to eye care for non-Indigenous, non-dominant ethnic groups in high-income countries: a scoping review protocol

Lisa Marie Hamm [1], Joanna Black [1], Helen Burn [2], Corina Grey [3], Matire Harwood [3], Roshini Peiris-John [3], Iris Gordon [2], Matthew J Burton [2,4], Jennifer R Evans [2], Jacqueline Ramke [1,2]

¹School of Optometry and Vision Science, The University of Auckland, Auckland, New Zealand
²International Centre for Eye Health, Faculty of Infectious and Tropical Diseases, London School of Hygiene and Tropical Medicine, London, UK
³School of Population Health, The University of Auckland, Auckland, New Zealand
⁴Moorfields Eye Hospital, London, UK

**Correspondence to**
Dr Jacqueline Ramke;
jacqueline.ramke@lshtm.ac.uk

## ABSTRACT

**Introduction** For many people, settling in a new country is associated with a new identity as an 'ethnic minority', one that can remain through future generations. People who are culturally distinct from the dominant population group may experience a variety of barriers to accessing healthcare, including linguistic and cultural barriers in communication, navigation of an unfamiliar health system and unconscious or overt discrimination. Here, we outline the protocol of a scoping review to identify, describe and summarise interventions aimed at improving access to eye care for non-Indigenous, non-dominant ethnic groups residing in high-income countries.

**Methods and analysis** We will search MEDLINE, Embase and Global Health from their inception to July 2019. We will include studies of any design that describe an intervention to promote access to eye care for non-Indigenous, non-dominant ethnic groups. Two authors will independently review titles, abstracts and full-text articles for inclusion. Reference lists from all included articles will also be searched. In cases of disagreement between initial reviewers, a third author will help resolve the conflict. For each included article, we will extract data about the target population, details of the intervention delivered and the effectiveness of or feedback from the intervention. Overall findings will be summarised with descriptive statistics and thematic analysis.

**Ethics and dissemination** This review will summarise existing literature and as such ethics approval is not required. We will publish the review in an open-access, peer-reviewed journal, and draft appropriate summaries for dissemination to the wider community. This wider community could include clinicians, policymakers, health service managers and organisations that work with non-dominant ethnic groups. Our findings will also feed into the ongoing Lancet Global Health Commission on Global Eye Health.

## INTRODUCTION
### Rationale

Equitable access to healthcare is critically important, but it is a challenge to both define and achieve.[1] Health systems are often implicitly structured to meet the needs and

### Strengths and limitations of this study

► This study will provide a comprehensive overview of the published literature on interventions to improve access to eye care for non-Indigenous, non-dominant ethnic groups in high-income countries.
► The review will be comprehensive, including published literature of all study designs, without time period or language restrictions.
► A potential limitation is that the population of interest can be difficult to define.
► Relevant evidence may exist in the grey literature, but our review is limited to the published literature.

preferences of members of the dominant group in any given population, which makes these systems more challenging for people with diverse backgrounds to navigate.[2 3] The axes of diversity vary widely (including socioeconomic status, gender, sexual orientation and Indigeneity) and are often intersectional. The challenges in navigation of healthcare systems are compounded for people with a non-dominant ethnic background, because the healthcare seeker is more likely to look, speak and communicate differently to their healthcare providers.[4]

The history of each ethnic group in a given place can influence how and to what extent health services strive to mitigate the vulnerabilities experienced by the group. For example, there is increasing recognition of overt institutional racism against colonised Indigenous populations and the impact this has on healthcare and health outcomes.[5] Formal efforts at restitution[6] have attempted to improve access to healthcare, such as government-funded services to rural and remote areas with high Indigenous populations, and health facilities within Indigenous communities. For this reason, we are

investigating service delivery models to improve access to eye care for Indigenous populations in a complementary scoping review,[7] and in the scoping review outlined here, we consider interventions to promote access to eye care for non-Indigenous, non-dominant ethnic groups.

A 'migrant' is a person who is living or has lived in a different place than they were born.[3] Using this definition, it was estimated that 3.4% of the global population (258 million people) were migrants in 2017.[8] People move away from their country of birth for a variety of reasons: many move for employment, others are forced from their home country because of civil unrest or violence, and some are moved through human trafficking and modern slavery.[3] Some migrants arrive in countries with a similar culture and language to their own, while others are faced with navigating a new cultural context, often finding themselves misunderstood or discriminated against and subject to many barriers to accessing quality healthcare.[3 4] These challenges can endure through future generations, with many people treated like perpetual foreigners in the only home they have known.

Non-dominant ethnic groups are vulnerable to poor access to healthcare in several ways. A lack of familiarity with local health systems or a fear that using community resources might compromise social acceptance, or immigration status can prevent people from seeking care.[9] When people from non-dominant ethnic groups do seek care, the healthcare provider is unlikely to share their native language or cultural heritage.[10] This can be associated with unarticulated differences in cultural beliefs about health[11] and a general breakdown in rapport or trust.[12] In the worst case, people are overtly disrespected in medical environments, compromising future health seeking behaviour.[10] Breakdowns in understanding are often magnified at a structural level. People from non-dominant ethnic groups often have limited power to impact the systems around them; they are less likely to be included in decision-making structures, or to be identified as a priority group for health funding.[3]

Similar issues impact eye care.[13 14] Studies from the USA report underutilisation of eye care services by non-dominant ethnic groups in general[15] and specifically by Latin Americans[16] and recent immigrants.[17] Although some public services are available (eg, Medicaid includes eye care and some school's vision screening programmes), the ability to fully use services is often compromised.[13 18 19]

Some interventions exist to promote access to quality eye care for vulnerable ethnic groups[17 20 21]; however, these studies are diverse in terms of the population targeted, the methodological framework and the eye problem addressed. Indeed, defining the target group is a challenge, given the difficulty in defining ethnicity and the overlap of challenges experienced due to ethnicity with those due to socioeconomic status, education, acculturation and geography. Given this diversity, a systematic review may be inconclusive at this time, and further primary studies would not adequately build on lessons learnt within this literature. A scoping review[22] appeared

the most appropriate method to map and summarise this field of research.[23] This protocol (and a parallel protocol[7]) were designed to inform a project to improve access to eye care services for Māori and Pacific people in Aotearoa (New Zealand), but the scope and implications are international.

## Definitions and concepts

The group of primary interest in this review is difficult to define. Self-identification of ethnicity is often fluid and nuanced,[24] and appropriate terminology within health research is actively debated.[25] Although 'ethnic minority' is commonly used in health research to refer to a group with a shared ethnic or cultural heritage, which differs from the dominant population where one resides, there is no accepted international definition.[6] For example, 'minority' can mean numerically smaller, or it can reference lack of power or dominance.[6] Some definitions include a will to preserve a cultural identity, while others note that group membership is involuntary or imposed (this distinction is sometimes captured in the differential use of 'ethnicity' as self-identity vs 'race' as an imposed identity).[25] Indeed, many terms related to the role of ethnicity in society carry different implicit meaning across countries and time. For example, terms like 'migrant', 'immigrant' and 'expatriate' each reflect a new identity when living in a new place, yet differential use can reveal assumptions about perceived wealth and influence. Similarly, the terms 'race', 'ethnicity', 'national' and 'visible minority' can carry nuanced assumptions that may not be shared internationally.

Within these challenges of terminology, we are interested in ethnic identities which are disempowering in their immediate context, and we refer to these groups as 'non-dominant ethnic groups'. This could include refugees and recent immigrants as well as those who have lived in the country of residence for many generations. Since we have chosen to address Indigenous populations in a separate review,[7] our definition here is limited to people who are not Indigenous to the country in which the study is located.

We have defined *eye care service delivery intervention* as any organised programme or activity designed to improve access to care, according to the patient-centred access to healthcare framework provided by Levesque *et al*.[26] Levesque's framework includes a progression from healthcare needs to perception of needs and desire for care, to healthcare seeking, reaching, utilisation and finally consequences (figure 1). The progression between stages depends on service factors, including acceptability, accessibility, availability and accommodation, affordability, and appropriateness, as well as the resources and knowledge of the patient, including their ability to perceive, seek, reach, pay for and engage with healthcare services.

The eye care that will be covered will include general services (prevention and treatment services, vision rehabilitation), as well as those for a particular condition or age group.

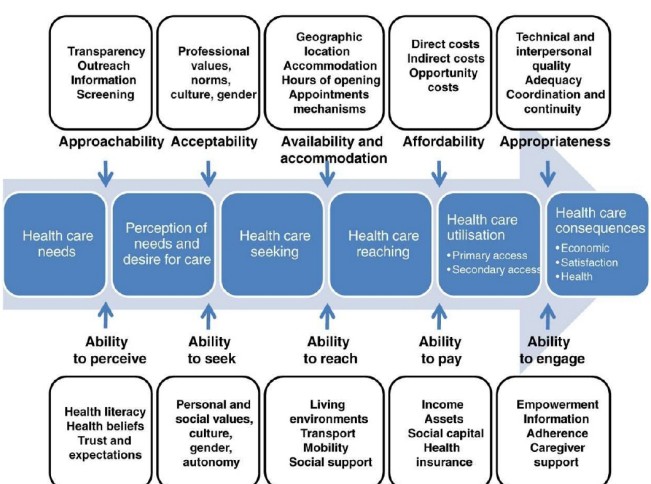

**Figure 1** Conceptual framework for access to healthcare (reproduced from Levesque et al[26]).

## METHODS AND ANALYSIS

We have reported this protocol in accordance with the relevant sections of the Preferred Reporting Items for Systematic Reviews and Meta-Analyses extension for Scoping Reviews (PRISMA-ScR) guideline.[27] The same guideline will be used to report the final review.

### Scoping review questions

We aim to answer the following three questions.

1. What is the extent of the published literature on interventions to promote access to eye care for non-Indigenous, non-dominant ethnic groups living in high-income countries?
2. What can we learn from reported effectiveness of interventions?
3. What can we learn from authors' reflections on the potential to improve on the interventions?

### Eligibility criteria

We are interested in studies that describe interventions to improve access to eye care services for non-dominant ethnic groups (as defined above) residing in high-income countries (as defined by the World Bank).[28] This includes recent migrants, refugees and those who have resided in the high-income country for multiple generations. Given the interplay between ethnicity and socioeconomic status,[29] interventions which could improve eye care services for ethnic minorities may have targeted another population (eg, 'urban poor'), but had a high proportion of participants from non-dominant ethnic groups. When these studies inform our objectives, we will include studies in which at least 50% of participants are from any non-dominant, non-Indigenous population group. Given the exploratory nature of a scoping review, we will iteratively discuss this component of the inclusion criteria with an aim to include the most relevant papers and will note any changes in the final review.

Studies which aim to improve provision of healthcare more generally may include eye care, for example,

interventions for diabetes care may include an assessment of diabetic retinopathy. In these cases, we will only include studies if there is sufficient detail on the eye care component of the intervention to be relevant as a stand-alone resource. We will exclude reviews, commentaries and editorials, but will check the reference list of review articles for potentially relevant studies. We will include all languages and study designs, but will exclude studies for which the full text is unavailable after exhausting university library resources.

### Search strategy

The authors collaborated to propose relevant search terms. Final terms and strategy (details in online supplementary file 1) were then refined for use within MEDLINE, Embase and Global Health databases by Cochrane Eyes and Vision's information specialist (IG). The literature will be searched to July 2019. An additional round of searching will be based on reference lists from included articles and relevant reviews. Due to resource constraints, we will not search the grey literature.

### Study selection

All the results from the search will be entered into Covidence (www.covidence.org) for screening. Two authors (from LMH, JR, JB, CG, RP-J or HB) will independently review each title and abstract and exclude those that do not meet the inclusion criteria. If the reviewers do not agree, the two reviewers will discuss and resolve. A third author will be consulted if no resolution can be found by the initial two reviewers. The full text of the selected articles will be reviewed, and the same two authors will independently vote to include or exclude articles. Again, conflict resolution will be handled by discussion, and a third reviewer if needed. A PRISMA flow diagram will be used to summarise the screening process.

### Data charting

A data extraction form will be developed based on the data items detailed below. The form will be piloted on five studies by each of LMH, JR, JB, CG and HB, and amendments made. Given the diversity of expected results, the charting process will remain iterative, with all changes to the data charting process noted. As with the screening process, two authors will independently chart each included article. The differences in charting will be resolved through discussion, and a third author called on to resolve discrepancies, if needed. If additional information is required from included studies, we will contact authors directly via email with a maximum of three attempts.

### Data items

The following data items will be collected during the data charting process:

1. Publication characteristics (title, year of publication, study design, country of origin and study setting).

2. Characteristics of the targeted group(s) (age, ethnicity, language, socioeconomic status and duration of residence in place of study).
3. Characteristics of the intervention (eye care context, targeted population's involvement in the development and implementation of the intervention, what was done to improve access to eye health and which dimensions of the Levesque framework for access were addressed and how).
4. Evaluated outcomes of the study (if the intervention was evaluated, how was it evaluated and what was the effectiveness, how many people were impacted by the intervention, how were baseline values and outcomes measured, and what analyses were used to draw conclusions).
5. Authors' reflections on the intervention (authors' reflections on what worked and why, what did not and why, and any suggestions for future interventions).

### Data synthesis

The interventions will be summarised descriptively and grouped according to the context, target population, eye condition(s) and access dimensions outlined above. Where interventions have been evaluated, the effectiveness, as well as identified strengths, weaknesses and suggested future directions will be summarised.

### ETHICS AND DISSEMINATION

This review will summarise existing literature and as such ethics approval is not required. We will publish the review in an open-access, peer-reviewed journal, and draft appropriate summaries for dissemination to the wider community. This wider community could include clinicians, policymakers, health service managers and organisations that work with non-dominant ethnic groups. Our findings will also feed into the ongoing Lancet Global Health Commission on Global Eye Health.[30]

### DISCUSSION

The challenges faced by some migrants in a new country can persist through many generations.[6] Access to public services, including eye care, is one such challenge.[14] The barriers are varied, influenced both by health system structures, leaders and workforce, and the resources and knowledge of the patient.[26] Given the diverse communities, with diverse barriers to eye care, varied interventions to improve access to eye care are likely to be needed. Here, we have outlined a protocol for a scoping review to summarise interventions to improve access to eye care for people from non-dominant ethnic groups.

We aim to map the available literature on the topic, which may take many forms. A scoping review lends itself well to this endeavour, especially given the anticipated diversity of the work in the field.[22] The scoping review outlined here is part of a larger study to improve access to eye care services for Indigenous and non-Indigenous

ethnic groups in Aotearoa (New Zealand). The findings will be useful to policymakers, health service managers and clinicians responsible for eye care services in New Zealand, as well as in other countries with similar marginalised population groups. In addition to publication in an open access journal, we will develop an accessible summary of the results for posting on institutional websites and dissemination at stakeholder meetings.

**Contributors** JR conceived the idea for the review. LMH and JR drafted and revised the protocol with suggestions from JB, HB, CG, MH, RP-J, MJB and JRE, who reviewed the protocol and provided feedback on the draft. IG constructed the search.

**Funding** This work was supported by The University of Auckland's Faculty Research Development Fund (grant number 3716758). JR is a Commonwealth Rutherford Fellow, funded by the UK government through the Commonwealth Scholarship Commission in the UK. LMH is the Robert Leitl Research Fellow, funded by the Robert Leitl Trust and The University of Auckland School of Optometry and Vision Science. CG holds a National Heart Foundation Fellowship. MJB is supported by the Wellcome Trust (207472/Z/17/Z).

**Competing interests** None declared.

**Patient and public involvement** Patients and/or the public were not involved in the design, or conduct, or reporting, or dissemination plans of this research.

**Patient consent for publication** Not required.

**Provenance and peer review** Not commissioned; externally peer reviewed.

**ORCID iDs**
Lisa Marie Hamm http://orcid.org/0000-0003-2777-7146
Joanna Black http://orcid.org/0000-0002-5100-8796
Helen Burn http://orcid.org/0000-0002-1469-8169
Corina Grey http://orcid.org/0000-0003-1751-1934
Matire Harwood http://orcid.org/0000-0003-1240-5139
Roshini Peiris-John http://orcid.org/0000-0001-7812-2268
Iris Gordon http://orcid.org/0000-0001-8143-8132
Matthew J Burton http://orcid.org/0000-0003-1872-9169
Jennifer R Evans http://orcid.org/0000-0002-6137-2030
Jacqueline Ramke http://orcid.org/0000-0002-5764-1306

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
