## [Reviewer comments · BMJ Open]

ARTICLE DETAILS

TITLE (PROVISIONAL)	Interventions to promote access to eye care for non-Indigenous, non-dominant ethnic groups in high-income countries: a scoping review protocol
AUTHORS	Hamm, Lisa; Black, Joanna; Burn, Helen; Grey, Corina; Harwood, Matire; Peiris-John, Roshini; Gordon, Iris; Burton, Matthew J; Evans, Jennifer; Ramke, Jacqueline

VERSION 1 – REVIEW

REVIEWER	Benoit Tousignant University of Montréal School of Optometry School of Public Health
REVIEW RETURNED	08-Jan-2020

GENERAL COMMENTS	This scoping review will be a very interesting and useful addition to the literature on access to eye health services to underserved populations. The manuscript is well written, with a clear and precise research question. The methodology is clear, complete and follows generally recognized methods for a scoping review (for more info see Colquhoun et al. Scoping reviews: time for clarity in definition, methods, and reporting. Journal of Clinical Epidemiology 67 (2014) 1291-1294). However, I would strongly recommend to add grey literature to the body of sources to search, especially for this type of clinical care models, which may easily be absent from indexed peer-reviewed literature. There is supporting literature for the use of grey literature in reviews (https://www.ncbi.nlm.nih.gov/pubmed/29266844) as well as some tools and sources from Cochrane for this: https://handbook-5-1.cochrane.org/chapter_6/6_2_1_8_grey_literature_databases.htm
---

REVIEWER	Dinesh Kaphle Queensland University of technology, Australia
REVIEW RETURNED	23-Feb-2020

GENERAL COMMENTS	It was my pleasure to review this manuscript, which I think is novel to this kind. I feel that the issue of health care access including eye care to non-indigenous, non-dominant ethnic group in high -income countries is often overlooked, although an advocacy to become a global citizen is getting more attention than ever. The manuscript acknowledges the fact that defining the study population is a challenge. It is good to make the search strategies quite extensive for this kind of research question where very little is known about it.
---

	Some minor comments are given below. Page 5 Scoping review questions- Key questions might be three not two. Page 6 Study selection- Two authors might be two out of six listed authors Page 5 Although Levesque et al. conceptual framework is presented in a figure in the previous publication (Burn et al.), it would be useful for readers to keep in this manuscript too.
--	--

REVIEWER	Dr Srinivas Marmamula L V Prasad Eye Institute, L V Prasad Marg, Banjara Hills, Hyderabad, India, 500034
REVIEW RETURNED	03-Mar-2020

GENERAL COMMENTS	Suggest adding another important question on the burden of vision loss in these groups as a background for this review. Also consider adding studies on barriers for utilization of eye care services in these groups, if available Not including studies where full text is not available may be reconsidered for review. Apart from published literature, the authors may also consider including case reports and narrative of the major NGO's working and supporting the initiatives in the region. Page 5, line 31: There are three questions listed and not two. Please correct it.
--

REVIEWER	Patricia McInerney University of the Witwatersrand South Africa
REVIEW RETURNED	17-Mar-2020

GENERAL COMMENTS	Thank you for the opportunity to review this protocol. I found the concepts underlying the need for scoping review very interesting. I just have two very minor comments to make: 1 - pg 5, it is stated that you aim to answer two questions, but list three. 2 - the searches are very comprehensive. have you run these searches? and if so, what number of potential papers did you arrive at?
---

VERSION 1 – AUTHOR RESPONSE

Reviewer: 1

This scoping review will be a very interesting and useful addition to the literature on access to eye health services to underserved populations.

The manuscript is well written, with a clear and precise research question. The methodology is clear, complete and follows generally recognized methods for a scoping review (for more info see Colquhoun et al. Scoping reviews: time for clarity in definition, methods, and reporting. Journal of Clinical Epidemiology 67 (2014) 1291-1294).

Thank you very much for your comments, and for highlighting this helpful resource.

However, I would strongly recommend to add grey literature to the body of sources to search, especially for this type of clinical care models, which may easily be absent from indexed peer-reviewed literature. There is supporting literature for the use of grey literature in reviews (<https://www.ncbi.nlm.nih.gov/pubmed/29266844>) as well as some tools and sources from Cochrane for this: https://handbook-5-1.cochrane.org/chapter_6/6_2_1_8_grey_literature_databases.htm

These are very helpful resources, thank you for highlighting them. We agree that resources from grey literature could be informative, however it was considered unfeasible for this project. During our initial search and screening phase, the volume of potentially relevant resources was extensive, which reinforced our decision to exclude grey literature of the nature you suggest. Rather than expanding the types of resources, we needed to make the difficult decision to further reduce our resources to exclude reviews, commentaries and editorials, and have edited the last sentence under the subheading 'Eligibility criteria' to reflect this.

"We will exclude reviews, commentaries and editorials, but will check the reference list of review articles for potentially relevant studies."

We also updated the last sentence of the section 'Search strategy'.

"An additional round of searching will be based on reference lists from included articles and relevant reviews."

In the strengths and limitations section, we have clarified that we are providing an overview of the published literature.

This study will provide a comprehensive overview of the published literature on interventions to improve access to eye care for non-Indigenous, non-dominant ethnic groups in high-income countries.

Reviewer: 2

It was my pleasure to review this manuscript, which I think is novel to this kind. I feel that the issue of health care access including eye care to non-indigenous, non-dominant ethnic group in high-income countries is often overlooked, although an advocacy to become a global citizen is getting more attention than ever.

The manuscript acknowledges the fact that defining the study population is a challenge. It is good to make the search strategies quite extensive for this kind of research question where very little is known about it.

Thank you very much for your comments.

Some minor comments are given below.

Page 5 Scoping review questions- Key questions might be three not two.

Thank you, fixed.

Page 6 Study selection- Two authors might be two out of six listed authors

Thank you for noting this point. We have revised the sentence as follows:

"Two authors (from LMH, JR, JB, CG, RPJ or HB) will independently"

Page 5 Although Levesque et al. conceptual framework is presented in a figure in the previous publication (Burn et al.), it would be useful for readers to keep in this manuscript too.

Thank you for this recommendation. We have added the figure.

Reviewer: 3

Suggest adding another important question on the burden of vision loss in these groups as a background for this review.

We appreciate this suggestion, and we will be including some of this information in the background of our subsequent paper. However, including a specific research question and proper analysis of the burden would require a different search and selection strategy.

Also consider adding studies on barriers for utilization of eye care services in these groups, if available

We agree that, like the burden, the barriers are important to articulate. However, as with burden, assessing barriers specifically would require a search and selection

strategy appropriate for that question. We have noted several such papers through our screening, and plan to briefly summarise this information in the introduction of our subsequent paper, rather than treating it as a separate research question.

Not including studies where full text is not available may be reconsidered for review.

So far, there is only one study we have not been able to find as we liaise with our extensive library services. We have revised the sentence as follows:

“We will include all languages and study designs, but will exclude studies for which the full text is unavailable after exhausting university library resources.”

Apart from published literature, the authors may also consider including case reports and narrative of the major NGO’s working and supporting the initiatives in the region.

Please see response to R1 who made a similar suggestion.

Page 5, line 31: There are three questions listed and not two. Please correct it.

Thank you, fixed.

Reviewer: 4

Thank you for the opportunity to review this protocol. I found the concepts underlying the need for scoping review very interesting.

Thank you for your interest.

I just have two very minor comments to make:

1 - pg 5, it is stated that you aim to answer two questions, but list three.

Thank you, fixed.

2 - the searches are very comprehensive. have you run these searches? and if so, what number of potential papers did you arrive at?

Since submitting the protocol for review we ran the search, which yielded 4912 items to screen.

VERSION 2 – REVIEW

REVIEWER	Benoit Tousignant University of Montreal School of Optometry, Canada Univerisyt of Montreal School of Public Health, Canada
REVIEW RETURNED	06-Apr-2020

GENERAL COMMENTS	I enjoyed reviewing the revision of this manuscript and appreciate the modifications performed. I reiterate that adding grey literature to the search would make this manuscript stronger and less susceptible to publication bias. For this type of clinical care models, many relevant examples may easily be absent from indexed peer-reviewed literature but rather be found in conference proceedings or in government agency reviews - both relevant and searchable sources of grey literature. There is supporting literature for the use of grey literature in reviews (https://www.ncbi.nlm.nih.gov/pubmed/29266844) as well as some tools and sources from Cochrane for this: https://handbook-5-1.cochrane.org/chapter_6/6_2_1_8_grey_literature_databases.htm . Should the authors choose not to include grey literature, a justification should be provided in the text.
---

VERSION 2 – AUTHOR RESPONSE

We acknowledge the benefits of searching the grey literature in the reviewer's feedback, but unfortunately are not in a position to expand our search at this time. Fortunately, our search of the published literature identified more than 70 studies for inclusion in our study.

We have added a justification for not searching the grey literature in the SEARCH STRATEGY section:

Due to resource constraints we will not search grey literature.